# CROSS-DOMAIN IMITATION LEARNING VIA OPTIMAL TRANSPORT

**Arnaud Fickinger**[13*]   **Samuel Cohen**[23]   **Stuart Russell**[1]   **Brandon Amos**[3]
[1]Berkeley AI Research   [2]University College London   [3]Facebook AI

## ABSTRACT

Cross-domain imitation learning studies how to leverage expert demonstrations of one agent to train an imitation agent with a different embodiment or morphology. Comparing trajectories and stationary distributions between the expert and imitation agents is challenging because they live on different systems that may not even have the same dimensionality. We propose *Gromov-Wasserstein Imitation Learning (GWIL)*, a method for cross-domain imitation that uses the Gromov-Wasserstein distance to align and compare states between the different spaces of the agents. Our theory formally characterizes the scenarios where GWIL preserves optimality, revealing its possibilities and limitations. We demonstrate the effectiveness of GWIL in non-trivial continuous control domains ranging from simple rigid transformation of the expert domain to arbitrary transformation of the state-action space. [1]

## 1  INTRODUCTION

Reinforcement learning (RL) methods have attained impressive results across a number of domains, e.g., Berner et al. (2019); Kober et al. (2013); Levine et al. (2016); Vinyals et al. (2019). However, the effectiveness of current RL method is heavily correlated to the quality of the training reward. Yet for many real-world tasks, designing dense and informative rewards require significant engineering effort. To alleviate this effort, imitation learning (IL) proposes to learn directly from expert demonstrations. Most current IL approaches can be applied solely to the simplest setting where the expert and the agent share the same embodiment and transition dynamics that live in the same state and action spaces. In particular, these approaches require expert demonstrations from the agent domain. Therefore, we might reconsider the utility of IL as it seems to only move the problem, from designing informative rewards to providing expert demonstrations, rather than solving it. However, if we relax the constraining setting of current IL methods, then natural imitation scenarios that genuinely alleviate engineering effort appear. Indeed, not requiring the same dynamics would enable agents to imitate humans and robots with different morphologies, hence widely enlarging the applicability of IL and alleviating the need for in-domain expert demonstrations.

This relaxed setting where the expert demonstrations comes from another domain has emerged as a budding area with more realistic assumptions (Gupta et al., 2017; Liu et al., 2019; Sermanet et al., 2018; Kim et al., 2020; Raychaudhuri et al., 2021) that we will refer to as *Cross-Domain Imitation Learning*. A common strategy of these works is to learn a mapping between the expert and agent domains. To do so, they require access to proxy tasks where both the expert and the agent act optimally in there respective domains. Under some structural assumptions, the learned map enables to transform a trajectory in the expert domain into the agent domain while preserving the optimality. Although these methods indeed relax the typical setting of IL, requiring proxy tasks heavily restrict the applicability of Cross-Domain IL. For example, it rules out imitating an expert never seen before as well as transferring to a new robot.

In this paper, we relax the assumptions of Cross-Domain IL and propose a benchmark and method that do not need access to proxy tasks. To do so, we depart from the point of view taken by previous work and formalize Cross-Domain IL as an optimal transport problem. We propose a method, that

---

*arnaud.fickinger@berkeley.edu, arnaudfickinger@fb.com

[1]Project site with videos and code: https://arnaudfickinger.github.io/gwil/

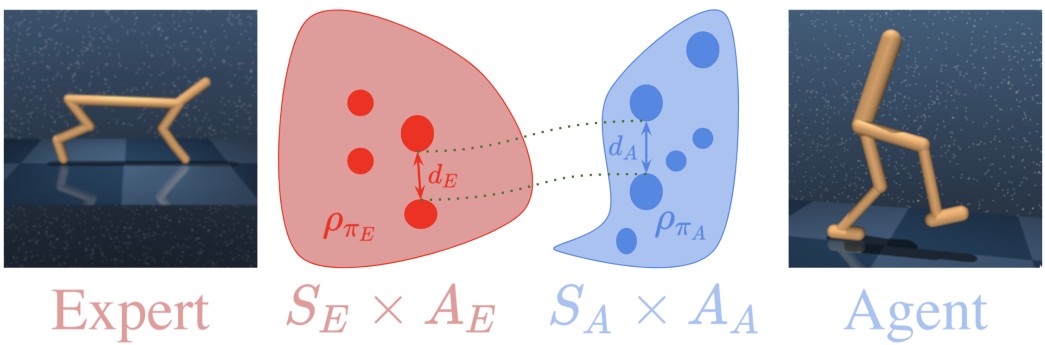

Figure 1: The Gromov-Wasserstein distance enables us to compare the stationary state-action distributions of two agents with different dynamics and state-action spaces. We use it as a pseudo-reward for cross-domain imitation learning.

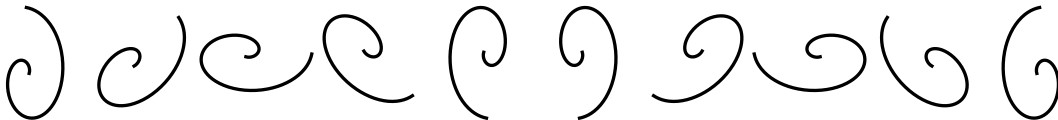

Figure 2: Isomorphic policies (definition 2) have the same pairwise distances within the state-action space of the stationary distributions. In Euclidean spaces, isometric transformations preserve these pairwise distances and include rotations, translations, and reflections.

we call *Gromov Wasserstein Imitation Learning (GWIL)*, that uses the Gromov-Wasserstein distance to solve the benchmark. We formally characterize the scenario where GWIL preserves optimality (theorem 1), revealing the possibilities and limitations. The construction of our proxy rewards to optimize optimal transport quantities using RL generalizes previous work that assumes uniform occupancy measures (Dadashi et al., 2020; Papagiannis & Li, 2020) and is of independent interest. Our experiments show that GWIL learns optimal behaviors with a single demonstration from another domain without any proxy tasks in non-trivial continuous control settings.

## 2   RELATED WORK

**Imitation learning.** An early approach to IL is Behavioral Cloning (Pomerleau, 1988; 1991) which amounts to training a classifier or regressor via supervised learning to replicate the expert's demonstration. Another key approach is Inverse Reinforcement Learning (Ng & Russell, 2000; Abbeel & Ng, 2004; Abbeel et al., 2010), which aims at learning a reward function under which the observed demonstration is optimal and can then be used to train a agent via RL. To bypass the need to learn the expert's reward function, Ho & Ermon (2016) show that IRL is a dual of an occupancy measure matching problem and propose an adversarial objective whose optimization approximately recover the expert's state-action occupancy measure, and a practical algorithm that uses a generative adversarial network (Goodfellow et al., 2014). While a number of recent work aims at improving this algorithm relative to the training instability caused by the minimax optimization, Primal Wasserstein Imitation Learning (PWIL) (Dadashi et al., 2020) and Sinkhorn Imitation Learning (SIL) (Papagiannis & Li, 2020) view IL as an optimal transport problem between occupancy measures to completely eliminate the minimax objective and outperforms adversarial methods in terms of sample efficiency. Heess et al. (2017); Peng et al. (2018); Zhu et al. (2018); Aytar et al. (2018) scale imitation learning to complex human-like locomotion and game behavior in non-trivial settings. Our work is an extension of Dadashi et al. (2020); Papagiannis & Li (2020) from the Wasserstein to the Gromov-Wasserstein setting. This takes us beyond limitation that the expert and imitator are in the same domain and into the cross-domain setting between agents that live in different spaces.

**Transfer learning across domains and morphologies.** Work transferring knowledge between different domains in RL typically learns a mapping between the state and action spaces. Ammar et al. (2015) use unsupervised manifold alignment to find a linear map between states that have similar

local geometry but assume access to hand-crafted features. More recent work in transfer learning across viewpoint and embodiment mismatch learn a state mapping without handcrafted features but assume access to paired and time-aligned demonstration from both domains (Gupta et al., 2017; Liu et al., 2018; Sermanet et al., 2018). Furthermore, Kim et al. (2020); Raychaudhuri et al. (2021) propose methods to learn a state mapping from unpaired and unaligned tasks. All these methods require proxy tasks, *i.e.* a set of pairs of expert demonstrations from both domains, which limit the applicability of these methods to real-world settings. Stadie et al. (2017) have proposed to combine adversarial learning and domain confusion to learn a policy in the agent's domain without proxy tasks but their method only works in the case of small viewpoint mismatch. Zakka et al. (2021) take a goal-driven perspective that seeks to imitate task progress rather than match fine-grained structural details to transfer between physical robots. In contrast, our method does not rely on learning an explicit cross-domain latent space between the agents, nor does it rely on proxy tasks. The Gromov-Wasserstein distance enables us to directly compare the different spaces without a shared space. The existing benchmark tasks we are aware of assume access to a set of demonstrations from *both* agents whereas the experiments in our paper *only* assume access to expert demonstrations. Finally, other domain adaptation and transfer learning settings use Gromov-Wasserstein variants, *e.g.* for transfer between word embedding spaces (Alvarez-Melis & Jaakkola, 2018) and image spaces (Vayer et al., 2020b).

## 3 PRELIMINARIES

**Metric Markov Decision Process.** An infinite-horizon discounted *Markov decision Process (MDP)* is a tuple $(S, A, R, P, p_0, \gamma)$ where $S$ and $A$ are state and action spaces, $P : S \times A \to \Delta(S)$ is the transition function, $R : S \times A \to \mathbb{R}$ is the reward function, $p_0 \in \Delta(S)$ is the initial state distribution and $\gamma$ is the discount factor. We equip MDPs with a distance $d : S \times A \to \mathbb{R}^+$ and call the tuple $(S, A, R, P, p_0, \gamma, d)$ a *metric MDP*.

**Gromov-Wasserstein distance.** Let $(\mathcal{X}, d_\mathcal{X}, \mu_\mathcal{X})$ and $(\mathcal{Y}, d_\mathcal{Y}, \mu_\mathcal{Y})$ be two metric measure spaces, where $d_\mathcal{X}, d_\mathcal{Y}$ are distances, and $\mu_\mathcal{X}, \mu_\mathcal{Y}$ are measures on their respective spaces[2]. Optimal transport (Villani, 2009; Peyré et al., 2019) studies how to compare measures. We will use the *Gromov-Wasserstein* distance (Mémoli, 2011) between metric measure spaces, which has been theoretically generalized and further studied in Sturm (2012); Peyré et al. (2016); Vayer (2020) and is defined by

$$\mathcal{GW}((\mathcal{X}, d_\mathcal{X}, \mu_\mathcal{X}), (\mathcal{Y}, d_\mathcal{Y}, \mu_\mathcal{Y}))^2 = \min_{u \in \mathcal{U}(\mu_\mathcal{X}, \mu_\mathcal{Y})} \sum_{\mathcal{X}^2 \times \mathcal{Y}^2} |d_\mathcal{X}(x, x') - d_\mathcal{Y}(y, y')|^2 u_{x,y} u_{x',y'}, \quad (1)$$

where $\mathcal{U}(\mu_\mathcal{X}, \mu_\mathcal{Y})$ is the set of couplings between the atoms of the measures defined by

$$\mathcal{U}(\mu_\mathcal{X}, \mu_\mathcal{Y}) = \left\{ u \in \mathbb{R}^{\mathcal{X} \times \mathcal{Y}} \middle| \forall x \in \mathcal{X}, \sum_{y \in \mathcal{Y}} u_{x,y} = \mu_\mathcal{X}(x), \forall y \in \mathcal{Y}, \sum_{x \in \mathcal{X}} u_{x,y} = \mu_\mathcal{Y}(y) \right\}.$$

$\mathcal{GW}$ compares the structure of two metric measure spaces by comparing the pairwise distances within each space to find the best isometry between the spaces. Figure 1 illustrates this distance in the case of the metric measure spaces $(S_E \times A_E, d_E, \rho_{\pi_E})$ and $(S_A \times A_A, d_A, \rho_{\pi_A})$.

## 4 CROSS-DOMAIN IMITATION LEARNING VIA OPTIMAL TRANSPORT

### 4.1 COMPARING POLICIES FROM ARBITRARILY DIFFERENT MDPS

For a stationary policy $\pi$ acting on a metric MDP $(S, A, R, P, \gamma, d)$, the *occupancy measure* is:

$$\rho_\pi : S \times A \to \mathbb{R} \qquad \rho(s, a) = \pi(a|s) \sum_{t=0}^{\infty} \gamma^t P(s_t = s|\pi).$$

We compare policies from arbitrarily different MDPs in terms of their occupancy measures.

---

[2]We use discrete spaces for readability but show empirical results in continuous spaces.

**Definition 1** (Gromov-Wasserstein distance between (isomorphic classes of) policies[3]). *Given an expert policy $\pi_E$ and an agent policy $\pi_A$ acting, respectively, on*

$$M_E = (S_E, A_E, R_E, P_E, T_E, d_E) \quad \text{and} \quad M_A = (S_A, A_A, R_A, P_A, T_A, d_A).$$

*We define the Gromov-Wasserstein distance between $\pi_E$ and $\pi_A$ as the Gromov-Wasserstein distance between the metric measure spaces $(S_E \times A_E, d_E, \rho_{\pi_E})$ and $(S_A \times A_A, d_A, \rho_{\pi_A})$[4]:*

$$\mathcal{GW}(\pi, \pi') = \mathcal{GW}((S_E \times A_E, d_E, \rho_{\pi_E}), (S_A \times A_A, d_A, \rho_{\pi_A})). \tag{2}$$

We now define isomorphisms between policies by comparing the state-action marginals and show that $\mathcal{GW}$ defines a distance between them. Figure 2 illustrates simple isomorphic policies.

**Definition 2** (Isomorphic policies). *Two policies $\pi_E$ and $\pi_A$ are isomorphic if there exists a bijection $\phi : \text{supp}[\rho_{\pi_E}] \to \text{supp}[\rho_{\pi_A}]$ that satisfies for all $(s_E, a_E), (s_{E'}, a_{E'}) \in \text{supp}[\rho_{\pi_E}]$ and $(s_A, a_A) \in \text{supp}[\rho_{\pi_A}]$:*

$$d_E((s_E, a_E), (s_{E'}, a_{E'})) = d_A(\phi(s_E, a_E), \phi(s_{E'}, a_{E'})) \tag{3}$$

$$\rho_{\pi_A}(s_A, a_A) = \rho_{\pi_E}(\phi^{-1}(s_E, a_E)) \tag{4}$$

*In other words, $\phi$ is an isometry between $(\text{supp}[\rho_{\pi_E}], d_E)$ and $(\text{supp}[\rho_{\pi_A}], d_A)$ and $\rho_{\pi_A}$ is the push-forward measure $\phi_\sharp(\rho_{\pi_E})$.*

**Proposition 1.** *$\mathcal{GW}$ defines a metric on the collection of all isomorphic classes of policies.*

*Proof.* By definition 1, $\mathcal{GW}(\pi_E, \pi_A) = 0$ if and only if $\mathcal{GW}((S_E, d_E, \rho_{\pi_E}), (S_A, d_A, \rho_{\pi_A})) = 0$. By Mémoli (2011, Theorem 5.1), this is true if and only if there is an isometry $\phi : \text{supp}[\rho_{\phi_E}] \to \text{supp}[\rho_{\phi_A}]$ such that $\rho_{\pi_A} = \phi_\sharp(\rho_{\pi_E})$. By definition 2, this is true if and only if $\pi_A$ and $\pi_E$ are isomorphic. The symmetry and triangle inequality follow from Mémoli (2011, Theorem 5.1). $\qquad \square$

The next theorem[5] gives a sufficient condition to recover, by minimizing $\mathcal{GW}$, an optimal policy[6] in the agent's domain up to an isometry.

**Theorem 1.** *Consider two MDPs*

$$M_E = (S_E, A_E, R_E, P_E, p_E, \gamma) \quad \text{and} \quad M_A = (S_A, A_A, R_A, P_A, p_A, \gamma).$$

*Suppose that there exists four distances $d_E^S, d_E^A, d_A^S, d_A^A$ defined on $S_E, A_E, S_A$ and $A_E$ respectively, and two isometries $\phi : (S_E, d_E^S) \to (S_A, d_A^S)$ and $\psi : (A_E, d_E^S) \to (A_S, d_A^S)$ such that for all $(s_E, a_E, s_E') \in S_E \times A_E \times S_E$ the three following conditions hold:*

$$R(s_E, a_E) = R_A(\phi(s_E), \psi(a_E)) \tag{5}$$

$$P_{E s_E, a_E}(s_E') = P_{A \phi(s_E)\psi(a_E)}(\phi(s_E')) \tag{6}$$

$$p_E(s_E) = p_A(\phi(s_E)). \tag{7}$$

*Consider an optimal policy $\pi_E^*$ in $M_E$. Suppose that $\pi_{GW}$ minimizes $\mathcal{GW}(\pi_E^*, \pi_{GW})$ with*

$$d_E : (s_E, a_E) \mapsto d_E^S(s_E) + d_E^A(a_E) \quad \text{and} \quad d_A : (s_A, a_A) \mapsto d_A^S(s_A) + d_A^A(a_A).$$

*Then $\pi_{GW}$ is isomorphic to an optimal policy in $M_A$.*

*Proof.* Consider the occupancy measure $\rho_A^* : S_A \times A_A \to \mathbb{R}$ given by

$$(s_A, a_A) \mapsto \rho_{\pi_E^*}(\phi^{-1}(s_A), \psi^{-1}(a_A)).$$

We first show that $\rho_A^*$ is feasible in $M_A$, *i.e.* there exists a policy $\pi_A^*$ acting in $M_A$ with occupancy measure $\rho_A^*$ (a). Then we show that $\pi_A^*$ is optimal in $M_A$ (b) and is isomorphic to $\pi_E^*$ (c). Finally we show that $\pi_{GW}$ is isomorphic to $\pi_A^*$, which concludes the proof (d).

---

[3]We later show that it is actually not a distance on policies but on isomorphic classes of policies.

[4]We always consider a policy in the context of the underlying metric MDP, such that every policy acting on $(S, A, R, P, T, d_E)$ are different from every policy acting on $(S, A, R, P, T, d_A)$ as soon as $d_E \neq d_A$. This guarantees that the Gromov-Wasserstein distance respects the identity of indiscernibles.

[5]Our proof is in finite state-action spaces for readability and can be directly extended to infinite spaces.

[6]A policy is optimal in the MDP $(S, A, R, P, \gamma, d)$ if it maximizes the expected return $\mathbb{E}\sum_{t=0}^\infty R(s_t, a_t)$.

(a) Consider $s_A \in S_A$. By definition of $\rho_A^*$,

$$\sum_{a_A \in A_A} \rho_A^*(s_A) = \sum_{a_A \in A_A} \rho_{\pi_E^*}(\phi^{-1}(s_A), \psi^{-1}(a_A)) = \sum_{a_E \in A_E} \rho_{\pi_E^*}(\phi^{-1}(s_A), a_E).$$

Since $\rho_{\pi_E^*}$ is feasible in $M$, it follows from Puterman (2014, Theorem 6.9.1) that

$$\sum_{a_E \in A_E} \rho_{\pi_E^*}(\phi^{-1}(s_A), a_E) = p_E(\phi^{-1}(s_A)) + \gamma \sum_{s_E \in S_E, a_E \in A_E} P_{E\, s_E, a_E}(\phi^{-1}(s_A)) + \rho_{\pi_E^*}(s_E, a_E).$$

By conditions 6 and 7 and by definition of $\rho_A^*$,

$$p_E(\phi^{-1}(s_A)) + \gamma \sum_{s_E \in S_E, a_E \in A_E} P_{E\, s_E, a_E}(\phi^{-1}(s_A)) + \rho_{\pi_E^*}(s_E, a_E)$$

$$= p_A(s_A) + \gamma \sum_{s_E \in S_E, a_E \in A_E} P_{A\, \phi(s_E), \psi(a_E)}(s_A) + \rho_A^*(\phi(s_E), \psi(a_E))$$

$$= p_A(s_A) + \gamma \sum_{s_A' \in S_A, a_A \in A_A} P_{A\, s_A', a_A}(s_A) + \rho_A^*(s_A', a_A).$$

It follows that

$$\sum_{a_A \in A_A} \rho_A^*(s_A) = p_A(s_A) + \gamma \sum_{s_A' \in S_A, a_A \in A_A} P_{A\, s_A', a_A}(s_A) + \rho_A^*(s_A', a_A).$$

Therefore, by Puterman (2014, Theorem 6.9.1), $\rho_A^*$ is feasible in $M_A$, *i.e.* there exists a policy $\pi_A^*$ acting in $M_A$ with occupancy measure $\rho_A^*$.

(b) By condition 7 and definition of $\rho_A^*$, the expected return of $\pi_A^*$ in $M_A$ is then

$$\sum_{s_A \in S_A, a_A \in A_A} \rho_A^*(s_A, a_A) R_A(s_A, a_A)$$

$$= \sum_{s_A \in S_A, a_A \in A_A} \rho_E^*(\phi^{-1}(s_A), \psi^{-1}(a_A)) R_E(\phi^{-1}(s_A), \psi^{-1}(a_A))$$

$$= \sum_{s_E \in S_E, a_E \in A_E} \rho_E^*(s_E, a_E) R_E(s_E, a_E)$$

Consider any policy $\pi_A$ in $M'$. By condition 7, the expected return of $\pi_A$ is

$$\sum_{s_A \in S_A, a_A \in A_A} \rho_{\pi_A}(s_A, a_A) R_A(s_A, a_A) = \sum_{s_E \in S_E, a_E \in A_E} \rho_{\pi_A}(\phi(s_E), \psi(a_E)) R_E(s_E, a_E).$$

Using the same arguments that we used to show that $\rho_A^*$ is feasible in $M'$, we can show that

$$(s_E, a_E) \mapsto \rho_{\pi_A}(\phi(s_E), \psi(a_E))$$

is feasible in $M$. It follows by optimality of $\pi_E^*$ in $M$ that

$$\sum_{s_E \in S_E, a_E \in A_E} \rho_{\pi_A}(\phi(s_E), \psi(a_E)) R_E(s_E, a_E) \leq \sum_{s_E \in S_E, a_E \in A_E} \rho_{\pi_E^*}(\phi(s_E), \psi(a_E)) R_E(s_E, a_E)$$

$$= \sum_{s_A \in S_A, a_A \in A_A} \rho_A^*(s_A, a_A) R_A(s_A, a_A).$$

It follows that $\pi_A^*$ is optimal in $M'$.

(c) Notice that

$$\xi : (s_E, a_E) \mapsto (\phi(s_E), \psi(a_E))$$

is an isometry between $(S_E \times A_E, d_E)$ and $(S_A \times A_A, d_A)$, where $d_E$ and $d_A$ and given, resp., by

$$(s_E, a_E) \mapsto d_E^S(s_E) + d_E^A(a_E) \quad \text{and} \quad (s_A, a_A) \mapsto d_A^S(s_A) + d_A^A(a_A).$$

Furthermore, by definition, $\rho_A^* = \xi_\sharp(\rho_E^*)$. Therefore by definition 2, $\pi_A^*$ is isomorphic to $\pi_E^*$.

---

**Algorithm 1** Gromov-Wasserstein imitation learning from a single expert demonstration.

**Inputs:** expert demonstration $\tau$, metrics on the expert ($d_E$) and agent ($d_A$) space
Initialize the imitation agent's policy $\pi_\theta$ and value estimates $V_\theta$
**while** Unconverged **do**
    Collect an episode $\tau'$
    Compute $\mathcal{GW}(\tau, \tau')$
    Set pseudo-rewards $r$ with eq. (9)
    Update $\pi_\theta$ and $V_\theta$ to optimize the pseudo-rewards
**end while**

---

(d) Recall from the statement of the theorem that $\pi_{GW}$ is a minimizer of $\mathcal{GW}(\pi_E^*, \pi_{GW})$. Since $\pi_A^*$ is isomorphic to $\pi_E^*$, it follows from prop. 1 that $\mathcal{GW}(\pi_E^*, \pi_A^*) = 0$. Therefore $\mathcal{GW}(\pi_E^*, \pi_{GW})$ must be 0. By prop. 1, it follows that there exists an isometry

$$\chi : (\text{supp}[\rho_E^*], d_E) \to (\text{supp}[\rho_{\pi_{GW}}], d_A)$$

such that $\rho_{\pi_{GW}} = \chi_\sharp(\rho_E^*)$. Notice that $\chi \circ \xi^{-1}|_{\text{supp}[\rho_A^*]}$ is an isometry from $(\text{supp}[\rho_A^*], d_A)$ to $(\text{supp}[\rho_{\pi_{GW}}], d_A)$ and $\rho_{\pi_{GW}} = (\chi \circ \xi^{-1}|_{\text{supp}[\rho_A^*]})_\sharp(\rho_A^*)$. It follows by definition 2 that $\pi_{GW}$ is isomorphic to $\pi_A^*$, an optimal policy in $M_A$, which concludes the proof. □

**Remark 1.** *Theorem 1 shows the possibilities and limitations of our method. It shows that our method can recover optimal policies even though arbitrary isometries are applied to the state and action spaces of the expert's domain. Importantly, we don't need to know the isometries, hence our method is applicable to a wide range of settings. We will show empirically that our method produces strong results in other settings where the environment are not isometric and don't even have the same dimension. However, a limitation of our method is that it recovers optimal policy only up to isometries. We will see that in practice, running our method on different seeds enables to find an optimal policy in the agent's domain.*

### 4.2 GROMOV-WASSERSTEIN IMITATION LEARNING

Minimizing $\mathcal{GW}$ between an expert and agent requires derivatives through the transition dynamics, which we typically don't have access to. We introduce a reward proxy suitable for training an agent's policy that minimizes $\mathcal{GW}$ via RL. Figure 1 illustrates the method. For readability, we combine expert state and action variables $(s_E, a_E)$ into single variables $z_E$, and similarly for agent state-action pairs. Also, we define $Z_E = S_E \times A_E$ and $Z_A = S_A \times A_A$.

**Definition 3.** *Given an expert policy $\pi_E$ and an agent policy $\pi_A$, the Gromov-Wasserstein reward of the agent is defined as $r_{\mathcal{GW}} : \text{supp}[\rho_{\pi_A}] \to \mathbb{R}$ given by*

$$r_{\mathcal{GW}}(z_A) = -\frac{1}{\rho_\pi(z_A)} \sum_{\substack{z_E \in Z_E \\ z_E' \in Z_E \\ z_A' \in Z_A}} |d_E(z_E, z_E')) - d_A(z_A, z_A')|^2 u_{z_E, z_A}^\star u_{z_E', z_A'}^\star$$

*where $u^\star$ is the coupling minimizing objective 1.*

**Proposition 2.** *If $\pi_A$ minimizes $\mathcal{GW}(\pi_E, \pi_A)$, then $\pi_A$ is an optimal policy for the reward $r_{\mathcal{GW}}$ as defined in definition 3.*

*Proof.* Suppose that $\pi_A$ minimizes $\mathcal{GW}(\pi_E, \pi_A)$, then by definition 1 $\pi_A$ maximizes

$$= -\sum_{\substack{z_E \in Z_E \\ z_E' \in Z_E \\ z_A \in Z_A \\ z_A' \in Z_A}} |d_E(z_E, z_E') - d_A(z_A, z_A')|^2 u_{z_A, z_E}^\star u_{z_A', z_E'}^\star$$

$$= -\sum_{z_A \in \text{supp}[\rho_{\pi_A}]} \frac{\rho_{\pi_A}(z_A)}{\rho_{\pi_A}(z_A)} \sum_{\substack{z_E \in Z_E \\ z_E' \in Z_E \\ z_A' \in Z_A}} |d_E(z_E, z_E') - d_A(z_A, z_A')|^2 u_{z_A, z_E}^\star u_{z_A', z_E'}^\star$$

Therefore, by Puterman (2014, Theorem 6.9.4), $\pi_A$ is an optimal policy for reward $r_{\mathcal{GW}}$. □

In practice we approximate the occupancy measures of $\pi$ by $\hat{\rho}_\pi(s, a) = \frac{1}{T}\sum_{t=1}^{T} \mathbb{1}(s = s_t \wedge a = a_t)$ where $\tau = (s_1, a_1, .., s_T, a_T)$ is a finite trajectory collected with $\pi$. Assuming that all state-action pairs in the trajectory are different[7], $\hat{\rho}$ is a uniform distribution. Given an expert trajectory $\tau_E$ and an agent trajectory $\tau_A$ [8], the (squared) Gromov-Wasserstein distance between the empirical occupancy measures is

$$\mathcal{GW}^2(\tau_E, \tau_A) = \min_{\theta \in \Theta^{T_E \times T_A}} \sum_{\substack{1 \le i, i' \le T_E \\ 1 \le j, j' \le T_A}} |d_E((s_i^E, a_i^E), (s_{i'}^E, a_{i'}^E)) - d_A((s_j^A, s_j^A), (s_{j'}^A, a_{j'}^A))|^2 \theta_{i,j}\theta_{i',j'}$$

(8)

where $\Theta$ is the set of is the set of couplings between the atoms of the uniform measures defined by

$$\Theta^{T \times T'} = \left\{ \theta \in \mathbb{R}^{T \times T'} \; \middle| \; \forall i \in [T], \sum_{j \in [T']} \theta_{i,j} = 1/T, \forall j \in [T'], \sum_{i \in [T]} \theta_{i,j} = 1/T' \right\}.$$

In this case the reward is given for every state-action pairs in the trajectory by:

$$r(s_j^A, s_j^A) = -T_A \sum_{\substack{1 \le i, i' \le T_E \\ 1 \le j' \le T_A}} |d_E((s_i^E, a_i^E), (s_{i'}^E, a_{i'}^E)) - d_A((s_j^A, s_j^A), (s_{j'}^A, a_{j'}^A))|^2 \theta_{i,j}^\star \theta_{i',j'}^\star$$

(9)

where $\theta^\star$ is the coupling minimizing objective 8.

In practice we drop the factor $T_A$ because it is the same for every state-action pairs in the trajectory.

**Remark 2.** *The construction of our reward proxy is defined for any occupancy measure and extends to previous work optimizing optimal transport quantities via RL that assumes uniform occupancy measure in the form of a trajectory to bypass the need for derivatives through the transition dynamics (Dadashi et al., 2020; Papagiannis & Li, 2020).*

**Computing the pseudo-rewards.** We compute the Gromov-Wasserstein distance using Peyré et al. (2016, Proposition 1) and its gradient using Peyré et al. (2016, Proposition 2). To compute the coupling minimizing 8, we use the conditional gradient method as in Ferradans et al. (2013).

**Optimizing the pseudo-rewards.** The pseudo-rewards we obtain from $\mathcal{GW}$ for the imitation agent enable us to turn the imitation learning problem into a reinforcement learning problem (Sutton & Barto, 2018) to find the optimal policy for the Markov decision process induced by the pseudo-rewards. We consider agents with continuous state-action spaces and thus do policy optimization with the soft actor-critic algorithm (Haarnoja et al., 2018). Algorithm 1 sums up GWIL in the case where a single expert trajectory is given to approximate the expert occupancy measure.

## 5 EXPERIMENTS

We propose a benchmark set for cross-domain IL methods consisting of 3 tasks and aiming at answering the following questions:

1. *Does GWIL recover optimal behaviors when the agent domain is a rigid transformation of the expert domain?* Yes, we demonstrate this with the maze in sect. 5.1.

2. *Can GWIL recover optimal behaviors when the agent has different state and action spaces than the expert?* Yes, we show in sect. 5.2 for *slightly* different state-action spaces between the cartpole and pendulum, and in sect. 5.3 for *significantly* different spaces between a walker and cheetah.

To answer these three questions, we use simulated continuous control tasks implemented in Mujoco (Todorov et al., 2012) and the DeepMind control suite (Tassa et al., 2018). We include videos of

---

[7]We can add the time step to the state to distinguish between two identical state-action pairs in the trajectory.

[8]Note that the Gromov-Wasserstein distance defined in equ. (6) does not depend on the temporal ordering of the trajectories.

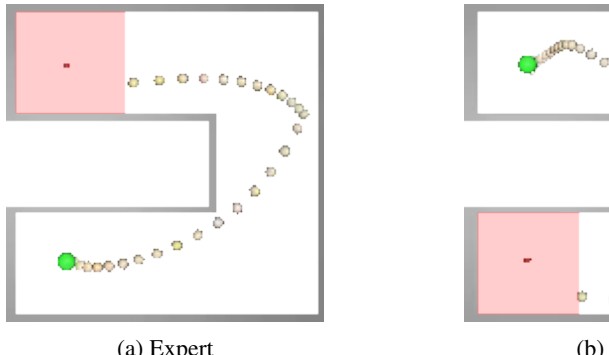

| (a) Expert | (b) Agent |

Figure 3: Given a single expert trajectory in the expert's domain (a), GWIL recovers an optimal policy in the agent's domain (b) without any external reward, as predicted by theorem 1. The green dot represents the initial state position and the episode ends when the agent reaches the goal represented by the red square.

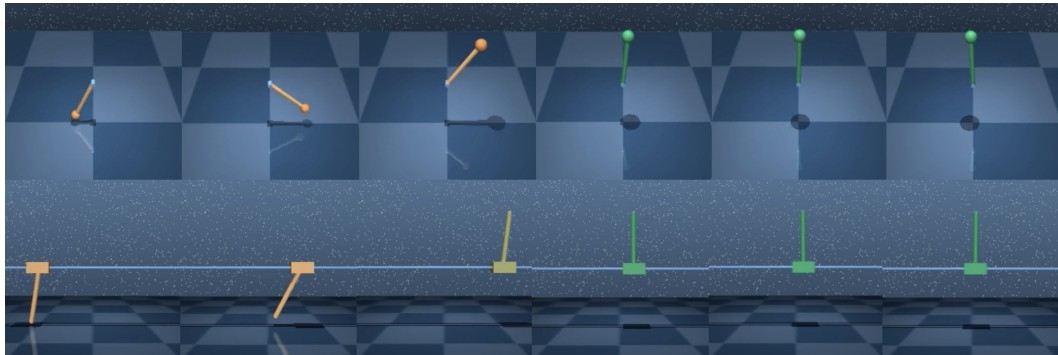

Figure 4: Given a single expert trajectory in the pendulum's domain (above), GWIL recovers the optimal behavior in the agent's domain (cartpole, below) without any external reward.

learned policies on our project site[9]. In all settings we use the Euclidean metric within the expert and agent spaces for $d_E$ and $d_A$.

## 5.1 AGENT DOMAIN IS A RIGID TRANSFORMATION OF THE EXPERT DOMAIN

We evaluate the capacity of IL methods to transfer to rigid transformation of the expert domain by using the PointMass Maze environment from Hejna et al. (2020). The agent's domain is obtained by applying a reflection to the expert's maze. This task satisfies the condition of theorem 1 with $\phi$ being the reflection through the central horizontal plan and $\psi$ being the reflection through the $x$-axis in the action space. Therefore by theorem 1, the agent's optimal policy should be isomorphic to the policy trained using GWIL. By looking at the geometry of the maze, it is clear that every policy in the isometry class of an optimal policy is optimal. Therefore we expect GWIL to recover an optimal policy in the agent's domain. Figure 3 shows that GWIL indeed recovers an optimal policy.

## 5.2 AGENT AND THE EXPERT HAVE SLIGHTLY DIFFERENT STATE AND ACTION SPACES

We evaluate here the capacity of IL methods to transfer to transformation that does not have to be rigid but description map should still be apparent by looking at the domains. A good example of such transformation is the one between the pendulum and cartpole. The pendulum is our expert's domain while cartpole constitutes our agent's domain. The expert is trained on the swingup task. Even though the transformation is not rigid, GWIL is able to recover the optimal behavior in the agent's domain as shown in fig. 4. Notice that pendulum and cartpole do not have the same state-action

---

[9]https://arnaudfickinger.github.io/gwil/

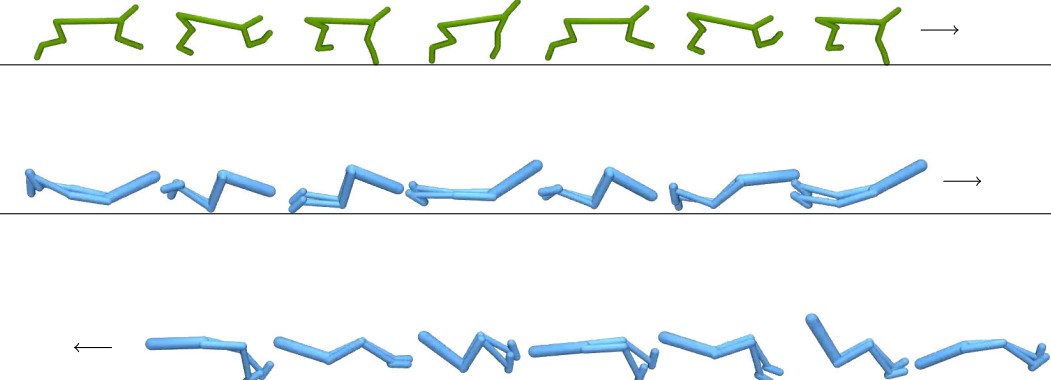

Figure 5: Given a single expert trajectory in the cheetah's domain (above), GWIL recovers the two elements of the optimal policy's isometry class in the agent's domain (walker), moving forward which is optimal (middle) and moving backward which is suboptimal (below). Interestingly, the resulting walker behaves like a cheetah.

space dimension: The pendulum has 3 dimensions while the cartpole has 5 dimensions. Therefore GWIL can indeed be applied to transfer between problems with different dimension.

### 5.3 AGENT AND THE EXPERT HAVE SIGNIFICANTLY DIFFERENT STATE AND ACTION SPACES

We evaluate here the capacity of IL methods to transfer to non-trivial transformation between domains. A good example of such transformation is two arbitrarily different morphologies from the DeepMind Control Suite such as the cheetah and walker. The cheetah constitutes our expert's domain while the walker constitutes our agent's domain. The expert is trained on the run task.

Although the mapping between these two domains is not trivial, minimizing the Gromov-Wasserstein solely enables the walker to interestingly learn to move backward and forward by imitating a cheetah. Since the isometry class of the optimal policy – moving forward– of the cheetah and walker contains a suboptimal element –moving backward–, we expect GWIL to recover one of these two trajectories. Indeed, depending on the seed used, GWIL produces a cheetah-imitating walker moving forward or a cheetah-imitating walker moving backward, as shown in fig. 5.

## 6 CONCLUSION

Our work demonstrates that optimal transport distances are a useful foundational tool for cross-domain imitation across incomparable spaces. Future directions include exploring:

1. **Scaling to more complex environments and agents** towards the goal of transferring the structure of many high-dimensional demonstrations of complex tasks into an agent.

2. The use of $\mathcal{GW}$ to help agents **explore in extremely sparse-reward environments** when we have expert demonstrations available from other agents.

3. How $\mathcal{GW}$ compares to **other optimal transport distances** that work apply between two metric MDPs, such as Alvarez-Melis et al. (2019), that have more flexibility over how the spaces are connected and what invariances the coupling has.

4. **Metrics aware of the MDP's temporal structure** such as Zhou & Torre (2009); Vayer et al. (2020a); Cohen et al. (2021) that build on dynamic time warping (Müller, 2007). The Gromov-Wasserstein ignores the temporal information and ordering present within the trajectories.

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

## A  OPTIMIZATION OF THE PROXY REWARD

In this section we show that the proxy reward introduced in equation 9 constitutes a learning signal that is easy to optimize using standards RL algorithms. Figure 6 shows proxy reward curves across 5 different seeds for the 3 environments. We observe that in each environment the SAC learner converges quickly and consistently to the asymptotic episodic return. Thus there is reason to think that the proxy reward introduced in equation 9 will be similarly easy to optimize in other cross-domain imitation settings.

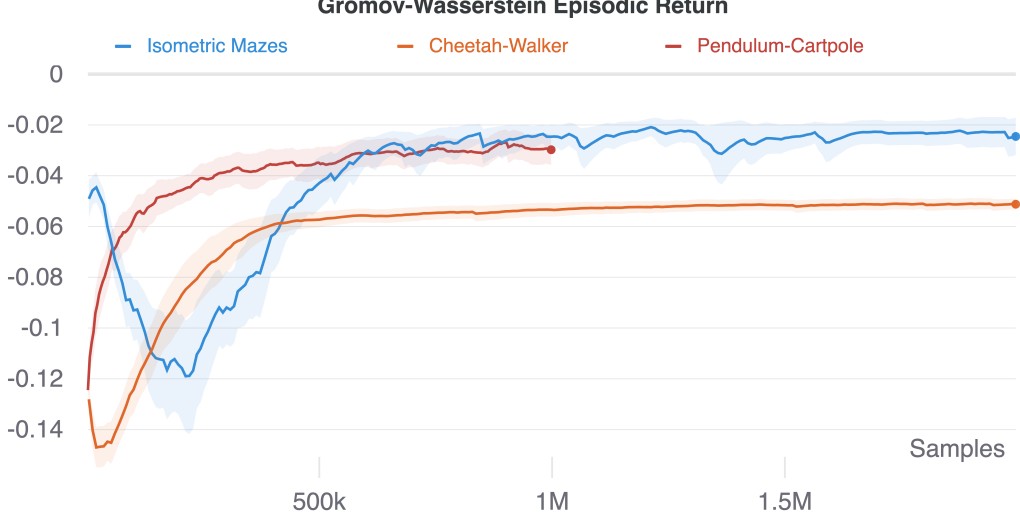

Figure 6: The proxy reward introduced in equation 9 gives a learning signal that is easily optimized using a standard RL algorithm.

## B  TRANSFER TO SPARSE-REWARD ENVIRONMENTS

In this section we show that GWIL can be used to facilitate learning in sparse-reward environments when the learner has only access to one expert demonstration from another domain. We compare GWIL to a baseline learner having access to a single demonstration from the same domain and minimizing the Wasserstein distance, as done in Dadashi et al. (2020). In these experiments, both agents are given a sparse reward signal in addition to their respective optimal transport proxy reward. We perform experiments in two sparse-reward environment. In the first environment, the agent controls a point mass in a maze and obtain a non-zero reward only if it reaches the end of the maze. In the second environment, which is a sparse version of cartpole, the agent controls a cartpole and obtains a non-zero reward only if he can maintain the cartpole up for 10 consecutive time steps. Note that a SAC agent fails to learn any meaningful behavior in both environments. Figure 7 shows that GWIL is competitive with the baseline learner in the sparse maze environment even though GWIL has only access to a demonstration from another domain, while the baseline learner has access to a demonstration from the same domain. Thus there is reason to think that GWIL efficiently and reliably extracts useful information from the expert domain and hence should work well in other cross-domain imitation settings.

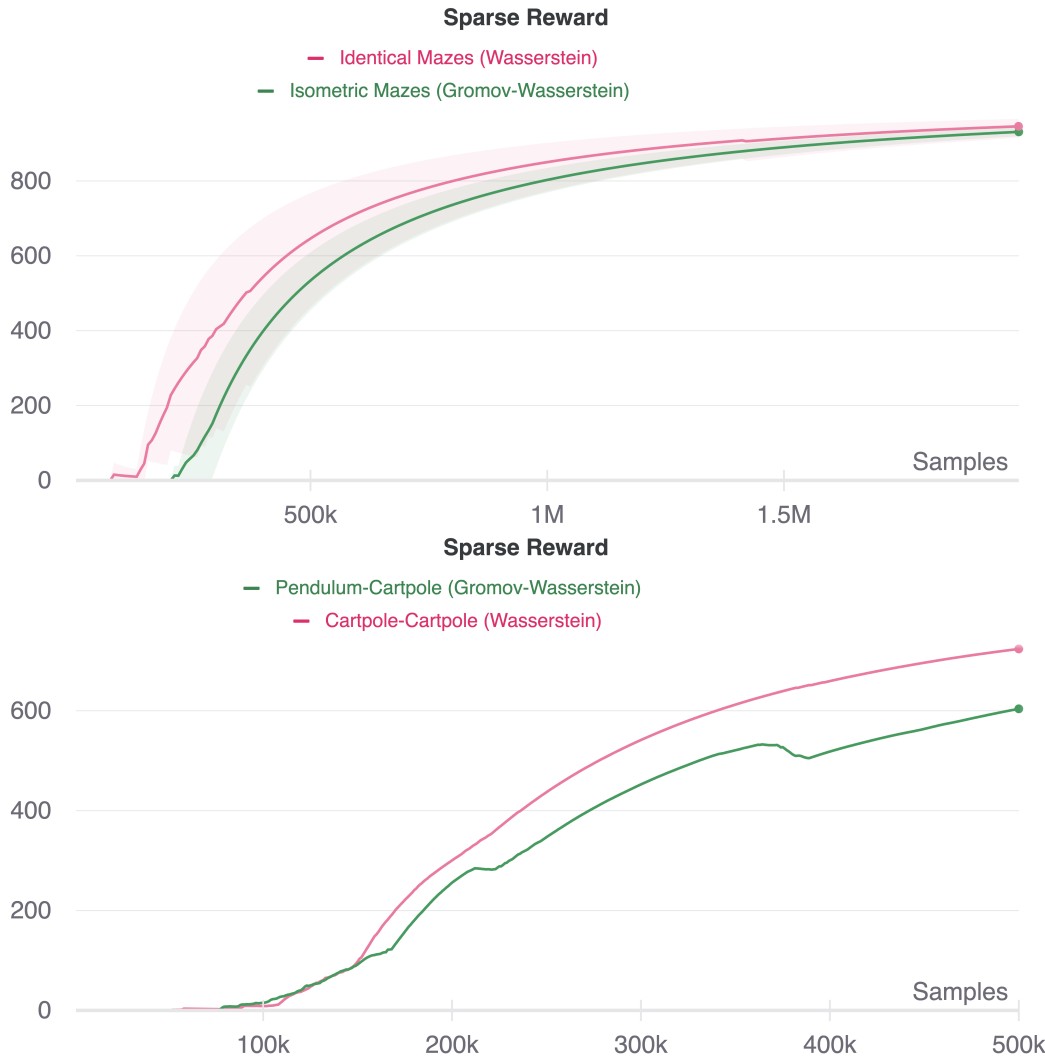

Figure 7: In sparse-reward environments, GWIL obtains similar performance than a baseline learner minimizing the Wasserstein distance to an expert in the same domain.

## C SCALABILITY OF GWIL

In this section we show that our implementation of GWIL offers good performance in terms of wall-clock time. Note that the bottleneck of our method is in the computation of the optimal coupling which only depends on the number of time steps in the trajectories, and not on the dimension of the expert and the agent. Hence our method naturally scales with the dimension of the problems. Furthermore, while we have not used any entropy regularizer in our experiments, entropy regularized methods have been introduced to enable Gromov-Wasserstein to scale to demanding machine learning tasks and can be easily incorporated into our code to further improve the scalability. Figure 8 compares the time taken by GWIL in the maze with the time taken by the baseline learner introduced in the previous section. It shows that imitating with Gromov-Wasserstein requires the same order of time than imitating with Wasserstein. Figure 9 compares the wall-clock time taken by a walker imitating a cheetah using GWIL to reach a walking speed (i.e., a horizontal velocity of 1) and the wall-clock time taken by a SAC walker trained to run. It shows that a GWIL walker imitating a cheetah reaches a walking speed faster than a SAC agent trained to run. Even though the SAC agent is optimizing for standing in addition to running, it was not obvious that GWIL could compete with SAC in terms of wall-clock time. These results gives hope that GWIL has the potential to scale to

more complex problems (possibly with an additional entropy regularizer) and be a useful way to learn by analogy.

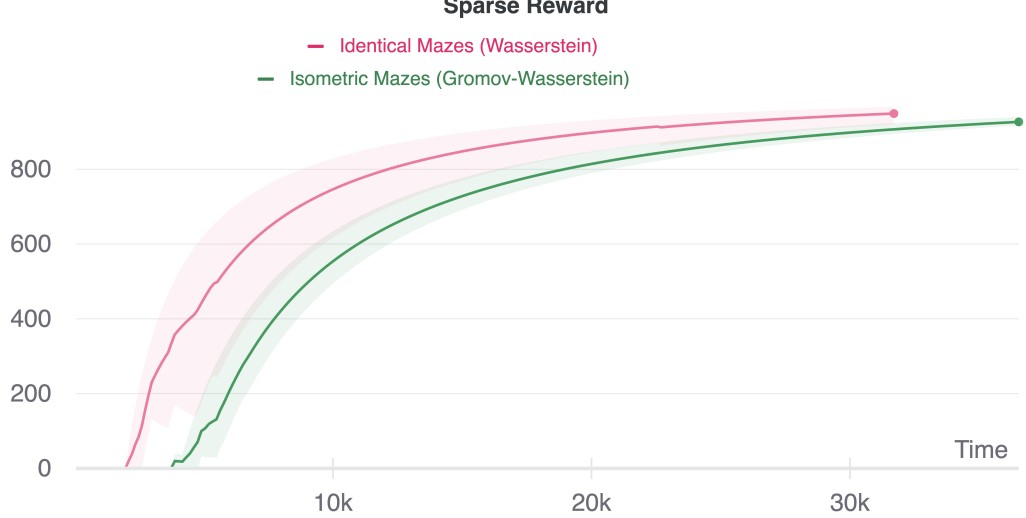

Figure 8: In the sparse maze environment, GWIL requieres the same order of wall-clock time than a baseline learner minimizing the Wasserstein distance to an expert in the same domain.

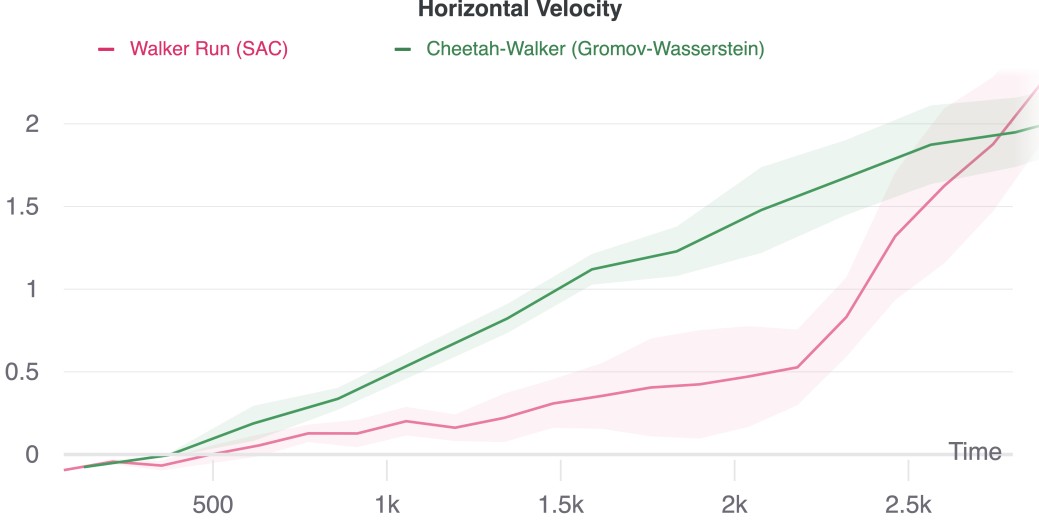

Figure 9: A GWIL walker imitating a cheetah reaches a walking speed faster than a SAC walker trained to run in terms of wall-clock time.

