# OpenReview forum: "Cross-Domain Imitation Learning via Optimal Transport"
_ICLR.cc/2022/Conference — ICLR 2022 Poster_

### Official Review · Reviewer_KcRn · 2021-11-01

**Correctness:** 3
**Technical Novelty And Significance:** 4
**Empirical Novelty And Significance:** 3
**Recommendation:** 8
**Confidence:** 4

**Main Review:**

Strengths:

This looks to be a strong contribution, attempting to solve an important problem. The paper is well-written and relatively easy to follow, and results and proofs are interesting.

Questions:

It's unclear how easy an objective the proxy reward is to maximise. I would appreciate more clarity around this (eg. can you show some reward curves across multiple seeds/ runs), and convergence. The Gromov Wasserstein distance is quite an expensive distance to compute, can you comment on the computational feasibility of optimising eq 7 and potential scaling issues?

Repeated mention of seed dependencies and effects is also concerning, I would appreciate more commentary on this.

While I agree there are certainly settings where the Gromov Wasserstein distance makes sense from an imitation learning perspective, recovery up to an isometry can be prohibitive. eg. A human showing a drone how to take-off could result in a policy that lands drones, which is the opposite of what was demonstrated. I would value some discussion on these limitations - would it make more sense to optimise a different distance metric, or to use a different eg. non-euclidean kernel in settings like these? I suspect this is a non-trivial choice, that needs a substantial level of domain specific knowledge  - does this then run counter the original objectives of this work?

Minor:

Pg 1. Intro is well written, but it would be great if Fig 2 could be shown earlier to give some more intuition into the Gromov Wasserstein distance and the solution framing.

Pg. 2 typo "This takes us beyond limitation..."

In existing imitation learning literature, much is made around limitations around learning from non-expert demonstrations. I would be interested to hear how the proposed approach would cope with these?

Eq 1 is in dire need of a figure to explain this.

As I understand it, although all proofs are provided in finite action and state spaces the proposed approach is said to scale to continuous spaces as ultimately it is only reliant on a suitable kernel function that can be expressed for continuous spaces. Is this correct?

"We will see that in practice, running our method on different seeds enables to find an optimal policy in the agent’s domain". How do we know which seed produced the "right" policy?

Fig 3 needs improving - it is extremely hard to see the agent.

Fig 4/5. I'd love to see videos of these policies - did it actually learn to balance cartpole, or just to swing up?






**Summary Of The Paper:**

This paper frames cross-domain imitation learning as an optimal transport problem using the Gromov-Wasserstein distance. This problem is highly relevant to imitation learning settings where there is often substantial domain mismatch between action and state spaces, eg. a humanoid robot learning to walk from a human demonstrator. The paper introduces a reward function that can be optimised and proves that this is equivalent to minimising the Gromov-Wasserstein distance between state action occupancies of an agent and expert. Substantial discussion/ proofs are included to show that minimising the Gromov-Wasserstein distance is equivalent to recovering an optimal policy up to an isometry. This is both a blessing and a curse, as it allows for optimal policies to be recovered under extreme changes in domain or differences, but does mean that recovered policies could be entirely unsuitable due to isometry.

The paper is well written and concisely written, although does get excessively mathy at times, when a figure could be more helpful. Experimental results corroborate the proofs and propositions, and highlight the value of the proposed approach.




**Summary Of The Review:**

I enjoyed reading this paper, and think it adds greatly to the conversation around cross-domain imitation learning. The proposed approach has a number of strengths and limitations which I would appreciate hearing more about, particularly when it comes to convergence speeds, repeatability and computational requirements, but also whether strengths/weaknesses of optimal recovery of a policy up to an isometry have just shifted the need for specification of a mapping between expert and agent into a different domain.

==== Post rebuttal comments ====
Thank you for engaging in the process, I still believe that this is a good paper.

---

> ### Author Response · Authors · 2021-11-18
> **Review Response**
>
> Thanks for your valuable feedback! We sum up below the additional experiments we ran and the paper revisions we made to address your concerns. Please let us know if our response addresses your concerns and if any issues remain.
>
> > It's unclear how easy an objective the proxy reward is to maximise. I would appreciate more clarity around this (eg. can you show some reward curves across multiple seeds/ runs), and convergence.
>
> Figure 6 shows that the proxy return converges quickly and consistently across all seeds and environments. Thus there is reason to think that our proxy reward will be similarly easy to optimize in other cross-domain imitation settings. Please take a look at the appendix in the revised version for further details.
>
> > The Gromov Wasserstein distance is quite an expensive distance to compute, can you comment on the computational feasibility of optimising eq 7 and potential scaling issues?
>
> The bottleneck of our method is in the computation of the optimal coupling which only depends on the number of time steps in the trajectories, and not on the dimension of the expert and the agent. Hence our method naturally scales with the dimension of the problems. Furthermore, while we have not used any entropy regularizer in our experiments, entropy regularized methods have been introduced to enable Gromov-Wasserstein to scale to demanding machine learning tasks and can be easily incorporated into our code to further improve the scalability.
> Furthermore, we ran two additional experiments that gives reason to think that GWIL offers a scalable approach to cross-domain imitation learning and can be applicable to a wide range of domains:
>
> - Figure 9 shows that a GWIL walker (30-dimensional robot) imitating a cheetah reaches a walking speed faster than a SAC walker trained to run in terms of wall-clock time.
> - Figure 8 shows that GWIL requires the same order of wall-clock time than a baseline learner minimizing the Wasserstein distance. Figure 9 shows that a GWIL walker imitating a cheetah reaches a walking speed faster than a SAC walker trained to run.
>
> Please take a look at the appendix in the revised version for further details.
>
> > I'd love to see videos of these policies - did it actually learn to balance cartpole, or just to swing up?
>
> We have provided a link to videos (https://sites.google.com/view/gwil/home) in the revised version. The first video shows that the GWIL agent actually learns to balance cartpole and not just to swing up.
>
> >Repeated mention of seed dependencies and effects is also concerning, I would appreciate more commentary on this. While I agree there are certainly settings where the Gromov Wasserstein distance makes sense from an imitation learning perspective, recovery up to an isometry can be prohibitive. [...] How do we know which seed produced the "right" policy?
>
> If we just care about obtaining an isometry, then we are consistently able to find them. If we have downstream objectives that we want the agent to attain, in addition to the isometries, we could imagine also augmenting the reward with them. This would influence the isometry to help achieve the objective. Additional experimental results presented in Figure 7 give reasons to think that GWIL is an efficient way to learn in sparse-reward environments. Please take a look at the appendix in the revised version for further details.
>
> > In existing imitation learning literature, much is made around limitations around learning from non-expert demonstrations. I would be interested to hear how the proposed approach would cope with these?
>
> It is an interesting question. At first sight, it seems that our method would be subject to the same limitations as imitation methods concerning non-expert trajectories, at least when the expert’s and the agent’s domains are the same. Whether our approach can abstract away the suboptimality of the non-expert when transferring to another domain, and to what extent it is due to our method versus the relation between the agent and expert domain, are interesting questions that need further investigation.
>
> > As I understand it, although all proofs are provided in finite action and state spaces the proposed approach is said to scale to continuous spaces as ultimately it is only reliant on a suitable kernel function that can be expressed for continuous spaces. Is this correct?
>
> It is a good remark. Theorem 1 can indeed be extended to the continuous case if we assume that the optimal transport quantities are well-defined in continuous space.
>
> > Fig 3 needs improving - it is extremely hard to see the agent.
>
> We have replaced Fig. 3 by two images representing the trajectories of the expert and the agent respectively, and it indeed greatly improves the clarity of the result.
>
> Finally, we have moved Fig 2 (now Fig 1 in the revised version) and it indeed enables us to provide some intuition when necessary.

---

> > ### Comment · Reviewer_KcRn · 2021-11-21
> > **Thanks**
> >
> > Thanks you for your detailed response to my questions, this is much appreciated.

---

### Official Review · Reviewer_ZsKi · 2021-11-01

**Correctness:** 3
**Technical Novelty And Significance:** 4
**Empirical Novelty And Significance:** 3
**Recommendation:** 6
**Confidence:** 3

**Main Review:**

Strengths:
- novel idea, to the best of my knowledge, for a difficult problem;  will inspire future work on "learning by analogy"
- combination of theory, and empirical experiments;  I'm surprised by the extent the method works in practice

Weaknesses:
- with no learning curves presented, it is unclear if the cross-domain imitation learning actually provides a benefit,
  for non-trivial systems, in terms of learning time or performance, as compared to learning from scratch.  It would be
  beneficial to see these learning curves, and a wall-clock compute time comparison.
- the limitations could be better articulated. The scalability is unclear (although it is unreasonable to expect the first
   iteration of an idea like this to scale right away). The isometry constraint is likely to be limiting in many settings,
  as is the choice of the Euclidean distance metric in the (state,action) space.
- lack of an intuitive presentation of the Gromov-Wasserstein distance. I had to go elsewhere to obtain the intuition.
  The actual method used to compute the GW distance, on discretely sampled trajectories.

The notation used for the GW sets-of-state action-pairs is confusing, i.e., \tau,
because the GW-distance is invariant with respect to temporal ordering (to my understanding), whereas the notation
GW(tau, tau')seems to imply that the ordering needs to be preserved.  Perhaps introduce a different notation for
the data when the temporal information no longer needs to be preserved?
The connections of the trajectories / (s,a) data / occupancy measures needs to be better articulated for this reader.

Is there a simple figure that could depict the essence of eqns 6 and 7?

Figure 1 appears to simply show rotations.  If the goal is to show isometry for translations,
it would be better to scatter the spirals more irregularly throughout the domain of the figure.
Similarly, why not include a reflection, as stated in the caption?

In Figure 3, the agents position is largely invisible.  Most of the readers may simply think there
is an editing mistake and that the same figure was included 8 times.

In Figure 4, do the top and bottom row come from GW-corresponding state/actions?

re: adding time to (s,a) to preserve uniqueness
Wouldn't this cause problems, given that the GW-distance would now include time?



**Summary Of The Paper:**

A method is proposed for cross-domain imitation learning, without resorting to any form of correspondence.
This is done using a Gromov-Wasserstein distance between policies (in practice, Euclidiean distances on collected state-action pairs,
within a given domain), which finds isometric transformations that best preserve distance measures, between the two domains.
Given an imitation domain and an expert domain with example trajectories, a pseudo-reward is computed
based on the degree to which the distances from a state to its neighbors in the imitation domain,
are preserved in the expert domain. Given these pseudo-rewards, as computed for collected episodes,
SAC is used as an RL algorithm to optimize the policy.
The paper contributes both a theoretical analysis and experiments with:  U-maze, pendulum-to-cart-and-pole, and half-cheetah-to-fallen-walker.


**Summary Of The Review:**

The paper introduces a novel idea for imitation learning.
It likely has many limitations, but the idea of find suitable imitation-based correspondences is one that is being
pursued on multiple fronts, and this is a new approach, with a mix of theory and some initial proof-of-concept examples.
The paper could do better at explaining core ideas, and still needs learning curves in order to understand the benefit of the
cross-domain transfer.

---

> ### Author Response · Authors · 2021-11-18
> **Review Response**
>
> Thanks for your valuable feedback! We sum up below the additional experiments we ran and the paper revisions we made to address your concerns. Please let us know if our response addresses your concerns and if any issues remain.
>
> > With no learning curves presented, it is unclear if the cross-domain imitation learning actually provides a benefit, for non-trivial systems, in terms of learning time or performance, as compared to learning from scratch. It would be beneficial to see these learning curves, and a wall-clock compute time comparison.
>
> This is a very interesting question and we provide additional experimental results that answer it in two different ways:
>
> - When learning from scratch is possible, we can indeed directly compare an agent learning by analogy to an agent learning from scratch. Fig 9 (revised version's appendix) shows that a GWIL walker (30-dimensional robot) imitating a cheetah reaches a walking speed faster than a SAC walker trained to run in terms of wall-clock time. Thus there is reason to think that learning by analogy using GWIL will provide a benefit over learning from scratch for other non-trivial systems as well.
> - When learning from scratch is not possible (i.e., sparse-reward environment), we have decided to compare the performance of our GWIL agent, learning by analogy, and a baseline agent learning by pure imitation. Figure 7 shows that GWIL obtains performances that are competitive with a baseline learner minimizing the Wasserstein distance, although the baseline learner has access to a demonstration from the same domain while GWIL has only access to a demonstration from a different domain. Thus there is reason to think that GWIL efficiently and reliably extracts useful information from the expert domain and hence is a good way to learn by analogy.
>
> >The scalability is unclear (although it is unreasonable to expect the first iteration of an idea like this to scale right away).
>
> The bottleneck of our method is in the computation of the optimal coupling which only depends on the number of time steps in the trajectories, and not on the dimension of the expert and the agent. Hence our method naturally scales with the dimension of the problems. Furthermore, while we have not used any entropy regularizer in our experiments, entropy regularized methods have been introduced to enable Gromov-Wasserstein to scale to demanding machine learning tasks and can be easily incorporated into our code to further improve the scalability.
>
> Furthermore, we ran two additional experiments that gives reason to think that GWIL offers a scalable approach to cross-domain imitation learning and can be applicable to a wide range of domains:
>
> - As mentioned above, Figure 9 shows that a GWIL walker (30-dimensional robot) imitating a cheetah reaches a walking speed faster than a SAC walker trained to run in terms of wall-clock time.
> - Figure 8 shows that GWIL requires the same order of wall-clock time as a baseline learner minimizing the Wasserstein distance. Figure 9 shows that a GWIL walker imitating a cheetah reaches a walking speed faster than a SAC walker trained to run.
>
> >The isometry constraint is likely to be limiting in many settings, as is the choice of the Euclidean distance metric in the (state,action) space.
>
> Our experiment shows that the isometry constraint in Theorem 1 is a sufficient condition but not a necessary condition. Furthermore, our experiments show that the Euclidean distance is sufficient to obtain non-trivial results, but other distances can be used directly with our code, which indeed increases the applicability of our method.
>
> > In Figure 4, do the top and bottom row come from GW-corresponding state/actions?
>
> They are screenshots that we manually picked from the trajectory videos. Please take a look at the videos (https://sites.google.com/view/gwil/home) for a better visualization of the recovered trajectories.
>
> > In Figure 3, the agents position is largely invisible. Most of the readers may simply think there is an editing mistake and that the same figure was included 8 times.
>
> We have replaced Fig. 3 by two images representing the trajectories of the expert and the agent respectively, and it indeed greatly improves the clarity of the result.
>
> > Adding time to (s,a) to preserve uniqueness Wouldn't this cause problems, given that the GW-distance would now include time?
>
> It is an interesting remark. If we add the time in the computation of the Euclidean distances in each domain, it might indeed enforce a temporal consistency that we don’t assume here. However, we are not likely to encounter two times the same state in continuous control domains and even if it’s the case, nothing prevents us from computing the Euclidean distances without the time and assuming that we have two different states when selecting the uniform distribution.
>
> Finally, we have changed Figure 1 (now Figure 2 in the revised version) and added a remark on the temporal-ordering invariance of expression 6.

---

> ### Author Response · Authors · 2021-11-20
> **Has our response addressed your concerns?**
>
> Hello reviewer ZsKi, we would be grateful if you can confirm whether our response has addressed your concerns, and let us know if any issues remain. We recap our response below.
>
> - To address your first concern about the lack of learning curves, we ran additional experiments. The additional experiments show that our agent, learning by analogy, offers a benefit over a SAC agent learning from scratch, and is competitive with a Wasserstein agent learning by pure imitation with access to a demonstration from the agent’s domain.
> - To address your second concern about the scalability, we explained why our method should scale to more complex problems and we ran additional experiments. The additional experiments show that our implementation has a reasonable wall-clock time efficiency compared to SAC and Wasserstein imitation.
> - To address your other concerns, we changed Figure 1 to include reflections, moved Figure 2 to give intuition about the metric sooner in the text, changed Figure 3 to clarify the result, added a link to videos to clarify Figure 4 and answered your question about adding the time to the state.
>
> Please take a look at our response, the appendix of the revised version and the videos for further details.

---

> > ### Comment · Reviewer_ZsKi · 2021-11-20
> > **comments on response**
> >
> > Thank you for the comments and additional experiments -- these are helpful!
> >
> > In terms of illustrating the intuition behind Gromov-Wasserstein, some ideas could also be drawn from the following diagram, which nicely contrasts GW against Wasserstein:  https://files.speakerdeck.com/presentations/cc7c19cffab64322a50c447fbf56a5c6/slide_40.jpg
> > There remains a semantic gap between the revised Figures 1 and 2, in that Figure 1 works on just a set of points, where as the continuous curve of Figure 2 implies temporal connectivity, which is there, but is ignored.  Thus introducing sample-points on top of two versions of the spirals in Figure 2 would make the connection more clear.  And I don't see the point of including 10 versions of the spiral just to illustrate translational/rotational/reflectional invariance; 2 or three versions (but with sample points) would be sufficient.
> >
> > Details about the actual implementation of the GW computations are missing.  Is there pseudocode or code about this important step? I'm not familiar with the setups needed for this, and other readers may in the same situation.
> >
> > Relatedly, in looking at the paper again, I can't find the actual RL algorithm that is used with the GW pseudo-rewards to do the actual policy learning. Currently there is an allusion to a policy and value function in Algo 1, but nothing further, unless I missed this somewhere?
> >
> > The Figure 9 comparison is good to have, but might be a bit misleading.  The basic walker2D environment doesn't allow for falling, so that is presumably true for the walker-run-SAC version, and not for the GW-version.  It could be that this graph simply captures the fact that walker2D finds "crawling" solutions more easily than running solutions.  Note that it doesn't really matter that much if the GW-version is SOTA or not -- I think that it has its own merits.  But any comparisons that are shown should be as fair as possible, or should have a description that provides necessary caveats.
> >
> > Thanks for creating the animations -- very useful.
> > However, I find the results to be difficult to interpret, as the "fallen walker" oscillates its body at roughly twice the frequency of the cheetah motion.   It is difficult to thus see this is a meaningful correspondence, given that the basic period of the overall motion is not preserved. On the other hand, because there is no temporal correspondence, perhaps the core take-away is that both are performing a cyclical motion?  Would simply matching to a set of points on circle produce the same result?  I'd be interested in seeing failure cases as well, as these provide as much insight as the successful cases.  Does GW-transfer across dissimilar tasks cause problems?  It could be that the answer is "no", in which case the GW transfer pseudo-rewards really don't provide much of a constraint.
> >
> > A candid discussion of limitations would be valuable, and is currently missing.
> > I.e., the only mentioned limitation (unless I have missed them; quite possible)  is that the optimal policy is only recovered up-to-isometries, which in some sense is actually a feature.
> > I do realize that frank discussions of limitations are unfortunately currently not a strong part of the culture of the ICLR community, and so I do not hold this paper to any higher standard on this than any other paper.  However, in the long run it would be healthy for the community to see papers do more in this direction.
> >
> > Thanks for any & all insights on the above points.

---

> > > ### Author Response · Authors · 2021-11-21
> > > **Additional response**
> > >
> > > Thank you for engaging in this discussion with us.
> > >
> > > > There remains a semantic gap between the revised Figures 1 and 2
> > >
> > > Introducing sample-points on top of the spirals is indeed a good way to connect both figures, we will add it to the paper.
> > >
> > > > Details about the actual implementation of the GW computations are missing.
> > >
> > > We compute the Gromov-Wasserstein distance using Peyré et al. (2016, Proposition 1) and its gradient using Peyré et al. (2016, Proposition 2). To compute the coupling minimizing 6, we use the conditional gradient method as discussed in Ferradans et al. (2013). We have added these details in paragraph "Computing the pseudo-rewards" page 7.
> > >
> > > > I can't find the actual RL algorithm that is used with the GW pseudo-rewards
> > >
> > > We state in paragraph "Optimizing the pseudo-rewards" page 7 that we use SAC to optimize the pseudo-rewards.
> > >
> > > > The Figure 9 comparison is good to have, but might be a bit misleading
> > >
> > > You are right, we have added more context to Figure 9 in Appendix C. Our hypothesis motivating this experiment was not "GWIL achieves the SOTA at Walker". Our hypothesis was "GWIL is not able to beat SAC in terms of wall-clock time even when SAC is optimizing an additional reward term". Indeed, it was not obvious to us that GWIL could compete with SAC in terms of wall-clock time. Figure 9 shows that our hypothesis is wrong, which is a good point for GWIL and gives hope that GWIL has the potential to scale to more complex problems (possibly with an additional entropy regularizer) and be a useful way to learn by analogy.
> > >
> > > >  It is difficult to thus see this is a meaningful correspondence, given that the basic period of the overall motion is not preserved.
> > >
> > > We will run the experiment again and visualize the coupling between both agents to have a better understanding of the correspondence.
> > >
> > > > I'd be interested in seeing failure cases as well, as these provide as much insight as the successful cases. [...] A candid discussion of limitations would be valuable, and is currently missing.
> > >
> > > We will run additional experiments to identify additional limitations of our method. Any idea you might have for additional limitations and experiments is very welcome.

---

> > > > ### Comment · Reviewer_ZsKi · 2021-11-21
> > > > **thanks for the quick reply**
> > > >
> > > > Thank you for the quick reply.
> > > > I had indeed missed the mention of soft actor critic as the algorithm, so thanks for pointing it out.
> > > > And thanks for adding the references for the GW-distance computations actually used in the implementation.
> > > > The biggest remaining insight would be if the paper could also describe how the method works when trying to transfer across a range of domains, ranging from those that are, intuitively, highly compatible, to others that are, intuitively, not compatible at all. If the paper could provide a basic understanding on this notion of compatibility, i.e., how good does the "analogy" have to be, then I'd be happy to bump up my score.
> > > > The author feedback thus far is truly appreciated.

---

> > > > > ### Author Response · Authors · 2021-11-30
> > > > > **Thanks for this discussion**
> > > > >
> > > > > The coupling experiments as well as the experiments with less obviously connected environments are not yet complete, but we will make sure to add the results to the final version!

---

### Official Review · Reviewer_tiAF · 2021-11-01

**Correctness:** 3
**Technical Novelty And Significance:** 3
**Empirical Novelty And Significance:** 3
**Recommendation:** 6
**Confidence:** 4

**Main Review:**

Strengths:

1.	This paper introduces and addresses an important and general cross domain imitation learning problem.
2.	Appling the Gromov-Wasserstein distance to align and compare two MDP domains provides insights to study cross domain imitation learning.
3.	The proposed GWIL is novel. The authors also well justified limitations theoretically.

Weakness:

1.	There are only 3 tasks shown in the experiment section. More experiment results are preferred to show the effectiveness of the proposed solution.
2.	Experiment results are not well-visualized. It would be better to give a link showing the results in animation.


**Summary Of The Paper:**

In this paper, the authors focus on a more general cross domain imitation learning problem where only expert demonstrations from one domain is available. To solve such a problem, the authors use the Gromov-Wasserstein distance to align and compare states between tasks from different domains and propose a Gromov-Wasserstein Imitation Learning (GWIL).  They also show theoretically the possibilities and limitations of GWIL.

**Summary Of The Review:**

In general, the paper addresses a general cross domain imitation learning problem where only expert demonstrations are available and proposes a novel GWIL. Though the experiment results are not visualized well, the work is highly likely to show new insights to researchers in imitation learning and domain adaptation domains.

---

> ### Author Response · Authors · 2021-11-18
> **Review Response**
>
> Thanks for your valuable feedback! We sum up below the additional experiments we ran and the paper revisions we made to address your concerns. Please let us know if our response addresses your concerns and if any issues remain.
>
> > Experiment results are not well-visualized. It would be better to give a link showing the results in animation.
>
> We have added a link to videos (https://sites.google.com/view/gwil/home) that show that the trajectories recovered by GWIL are smooth and meaningful in terms of the downstream task, which will hopefully further convince you of the effectiveness of the proposed solution:
>
> - The first video shows that GWIL actually learns to balance the cartpole, and not just to swing up as it was suspected by reviewer KcRn.
> - The second video shows that GWIL can efficiently transfer from a 23-dimensional robot (cheetah) to a 30-dimensional robot (walker), producing a trajectory that appears smooth and with an appropriate speed.
> - The third video shows that GWIL produces a trajectory that appears smooth in the maze environment.
>
> Please take a look at the website (https://sites.google.com/view/gwil/home) for further details.
>
> > More experiment results are preferred to show the effectiveness of the proposed solution.
>
> We have added experiment results to our paper supporting the three following claims concerning the effectiveness of the proposed solution:
>
> - **The learning signal provided by our proxy reward is easy to optimize using standard RL algorithms** (Appendix A): Figure 6 shows that the proxy return converges quickly and consistently across all seeds and environments. Thus there is reason to think that our proxy reward will be similarly easy to optimize in other cross-domain imitation settings.
> - **GWIL is an efficient way to learn in sparse-reward environments when only demonstrations from another domain is available** (Appendix B): Figure 7 shows that GWIL obtains performances that are competitive with a baseline learner minimizing the Wasserstein distance, although the baseline learner has access to a demonstration from the same domain while GWIL has only access to a demonstration from a different domain. Thus there is reason to think that GWIL efficiently and reliably extracts useful information from the expert domain and hence should work well in other cross-domain imitation settings.
> - **Our current implementation of GWIL offers good performance in terms of wall-clock time efficiency** (Appendix C): Figure 8 shows that GWIL requires the same order of wall-clock time than a baseline learner minimizing the Wasserstein distance. Figure 9 shows that a GWIL walker imitating a cheetah reaches a walking speed faster than a SAC walker trained to run in terms of wall-clock time. Thus there is reason to think that GWIL offers a scalable approach to cross-domain imitation learning and can be applicable to a wide range of domains.
>
> Please take a look at the appendix in the revised version of the paper for further details.

---

> ### Author Response · Authors · 2021-11-20
> **Has our response addressed your concerns?**
>
> Hello reviewer tiAF, we would be grateful if you can confirm whether our response has addressed your concerns, and let us know if any issues remain. We recap our response below.
>
> - To address your first concern about the lack of experiments, we ran 5 additional experiments. The additional experiments show that our method is easy to optimize, enables efficient transfer in sparse environments and scales well.
> - To address your second concern about the clarity of the results, we added a link to videos of the trajectories. The videos show that the trajectories recovered by GWIL are smooth and meaningful in terms of the downstream task.
>
> Please take a look at our response, the appendix of the revised version and the videos for further details.

---

> > ### Comment · Reviewer_tiAF · 2021-11-22
> > **Thank you for the detailed response**
> >
> > Thanks you for the detailed response, which makes sense to me.

---

### Official Review · Reviewer_pWm6 · 2021-11-03

**Correctness:** 3
**Technical Novelty And Significance:** 3
**Empirical Novelty And Significance:** 3
**Recommendation:** 6
**Confidence:** 4

**Main Review:**

Strengths:
- Good mathematical grounding
- Further exploration of an interesting alternative to map optimal policies to new embodiments (possible practical applications)
- Well written

Weaknesses:
- The main issue with this paper is the experimental evaluation. The presented results are just images of three cases. The images of the first case (Fig. 3) are hard to see. There is no numerical performance (success, reward, some degree of progress). This makes the experimental evaluation insufficient to understand the applicability of the presented approach. Add more experiments, numerical performance, different metrics…


**Summary Of The Paper:**

This paper presents a method to transfer policies between different MDPs based on the minimization of Gromov-Wasserstein distance. This distance provides a pseudo-reward that can be used to learn via RL the optimal policy in the target MDP given an optimal policy in the original MDP. The method is optimal if the MDPs can be mapped into each other through an isometry, but works also empirically in other cases.


**Summary Of The Review:**

In summary, while the paper presents an interesting turn on previously presented ideas, and the mathematical foundation is well worked out, the experimental evaluation is insufficient to support conclusions about the method.

---

> ### Author Response · Authors · 2021-11-18
> **Review Response**
>
> Thanks for your valuable feedback! We sum up below the additional experiments we ran and the paper revisions we made to address your concerns. Please let us know if our response addresses your concerns and if any issues remain.
>
> > There is no numerical performance (success, reward, some degree of progress). This makes the experimental evaluation insufficient to understand the applicability of the presented approach.
>
> We have added numerical performances to our paper supporting the three following claims concerning the applicability of the presented approach:
>
> - **The learning signal provided by our proxy reward is easy to optimize using standard RL algorithms** (Appendix A): Figure 6 shows that the proxy return converges quickly and consistently across all seeds and environments. Thus there is reason to think that our proxy reward will be similarly easy to optimize in other cross-domain imitation settings.
> - **GWIL is an efficient way to learn in sparse-reward environments when only demonstrations from another domain is available** (Appendix B): Figure 7 shows that GWIL obtains performances that are competitive with a baseline learner minimizing the Wasserstein distance, although the baseline learner has access to a demonstration from the same domain while GWIL has only access to a demonstration from a different domain. Thus there is reason to think that GWIL efficiently and reliably extracts useful information from the expert domain and hence should work well in other cross-domain imitation settings.
> - **Our current implementation of GWIL offers good performance in terms of wall-clock time efficiency** (Appendix C): Figure 8 shows that GWIL requires the same order of wall-clock time than a baseline learner minimizing the Wasserstein distance. Figure 9 shows that a GWIL walker imitating a cheetah reaches a walking speed faster than a SAC walker trained to run in terms of wall-clock time. Thus there is reason to think that GWIL offers a scalable approach to cross-domain imitation learning and can be applicable to a wide range of domains.
>
> Please take a look at the appendix in the revised version of the paper for further details.
>
> Furthermore, we have added a link to videos (https://sites.google.com/view/gwil/home) that show that the trajectories recovered by GWIL are smooth and meaningful in terms of the downstream task, which will hopefully further convince you of the applicability of the presented approach:
>
> - The first video shows that GWIL actually learns to balance the cartpole, and not just to swing up as it was suspected by reviewer KcRn.
> - The second video shows that GWIL can efficiently transfer from a 23-dimensional robot (cheetah) to a 30-dimensional robot (walker), producing a trajectory that appears smooth and with an appropriate speed.
> - The third video shows that GWIL produces a trajectory that appears smooth in the maze environment.
>
> Please take a look at the website (https://sites.google.com/view/gwil/home) for further details.
>
> > The images of the first case (Fig. 3) are hard to see.
>
> We have replaced Fig. 3 by two images representing the trajectories of the expert and the agent respectively, and it indeed greatly improves the clarity of the result.

---

> > ### Comment · Reviewer_pWm6 · 2021-11-22
> > **Post rebuttal**
> >
> > Thank you for your responses. I updated my score

---

> ### Author Response · Authors · 2021-11-20
> **Has our response addressed your concerns?**
>
> Hello reviewer pWm6, we would be grateful if you can confirm whether our response has addressed your concerns, and let us know if any issues remain. To address your concern about the empirical evaluation, we:
>
> - ran 5 additional experiments with numerical performance showing that our method is easy to optimize, enables efficient transfer in sparse environments and scales well,
> - added a link to videos showing that the trajectories recovered by GWIL are smooth and meaningful in terms of the downstream task,
> - changed figure 3 to improve the clarity of the result.
>
> Please take a look at our response, the appendix of the revised version and the videos for further details.

---

### Decision · Program_Chairs · 2022-01-20

**Decision:**

Accept (Poster)

**Comment:**

All reviewers suggested acceptance of the paper based on that the paper addresses an important problem and presents and validates interesting ideas for approaching it. Therea are some concerns regarding limited experiments - I'd like to encourage the authors to make an effort to address these concerns and also a few others raised in the reviews in the final version of their paper. The authors already made several updates to their paper in that regard during the discussion phase so I believe that the paper would be an interesting conttribution to the conference and I am recommending acceptance of the paper.